## [Decision Letter · Decision Letter 0]

24 Dec 2025

PGENETICS-D-25-01242

The genetic basis of mimicry in the snowy bumble bee (B. niveatus) in Anatolia with insights from a color polymorphic gynandromorph

PLOS Genetics

Dear Dr. Hines,

Thank you for submitting your manuscript to PLOS Genetics. After careful consideration, we feel that it has merit but does not fully meet PLOS Genetics's publication criteria as it currently stands. Therefore, we invite you to submit a revised version of the manuscript that addresses the points raised during the review process.

The following recommendations seem particularly important:

Please establish a clear taxonomic nomenclature (whether each name refers to a species, a subspecies, or a color morph), and follow it consistently throughout the paper.Confirm that the causative gene is the true ortholog of dipteran Bar-H2 (explain when the duplication occurred, which paralogs are present in Hymenoptera and show a small phylogeny confirming that the Bombus and Drosophila Bar-H2 genes are orthologous).It does seem plausible that one of the alleles may contain sequences that are absent in the reference genome (a structural variant); such sequences would not be detected by standard GWAS and would not be assembled via reference mapping.  A kmer-based GWAS, as suggested by one of the reviewers, would detect such a variant and allow each allele to be assembled independently of the reference genome and then compared.  Potentially, a structural variant may not even be immediately adjacent to the duplication, as long as it’s in strong LD with it.  If you do not want to perform a kmer-based GWAS, please add appropriate caveats.  And at a minimum, a more detailed presentation of Sanger sequencing and alignment is necessary to illustrate the duplication and the association of the duplicated and non-duplicated alleles with color polymorphism.Gene/allele nomenclature is highly organism-specific so it’s not for the editor to decide.  I can only encourage the authors to design a clear, easy to interpret genetic nomenclature that will remain consistent in all future work on this group of species.

We look forward to receiving your revised manuscript.

Kind regards,

Artyom Kopp

Academic Editor

PLOS Genetics

Justin Fay

Section Editor

PLOS Genetics

Aimée Dudley

Editor-in-Chief

PLOS Genetics

Anne Goriely

Editor-in-Chief

PLOS Genetics

**Journal Requirements:**

At this stage, the following Authors/Authors require contributions: Tunç Dabak, Çiğdem Özenirler, Ece Kamalak, Cecil Smith, Seçil Aytekin, Ahmet Murat Aytekin, and Heather M. Hines. Please ensure that the full contributions of each author are acknowledged in the "Add/Edit/Remove Authors" section of our submission form.

The list of CRediT author contributions may be found here: https://journals.plos.org/plosgenetics/s/authorship#loc-author-contributions

2) We have noticed that you have uploaded Supporting Information files, but you have not included a list of legends. Please add a full list of legends for your Supporting Information files after the references list.

3) Some material included in your submission may be copyrighted. According to PLOSu2019s copyright policy, authors who use figures or other material (e.g., graphics, clipart, maps) from another author or copyright holder must demonstrate or obtain permission to publish this material under the Creative Commons Attribution 4.0 International (CC BY 4.0) License used by PLOS journals. Please closely review the details of PLOSu2019s copyright requirements here: PLOS Licenses and Copyright. If you need to request permissions from a copyright holder, you may use PLOS's Copyright Content Permission form.

Potential Copyright Issues:

i) Please confirm (a) that you are the photographer of S1, S2, 1, and 5, or (b) provide written permission from the photographer to publish the photo(s) under our CC BY 4.0 license.

ii) Figures 1, and 2. Please confirm whether you drew the images / clip-art within the figure panels by hand. If you did not draw the images, please provide (a) a link to the source of the images or icons and their license / terms of use; or (b) written permission from the copyright holder to publish the images or icons under our CC BY 4.0 license. Alternatively, you may replace the images with open source alternatives. See these open source resources you may use to replace images / clip-art:

iii) Figures 1, and 2. Please (a) provide a direct link to the base layer of the map (i.e., the country or region border shape) and ensure this is also included in the figure legend; and (b) provide a link to the terms of use / license information for the base layer image or shapefile. We cannot publish proprietary or copyrighted maps (e.g. Google Maps, Mapquest) and the terms of use for your map base layer must be compatible with our CC BY 4.0 license.

4) Please amend your detailed Financial Disclosure statement. This is published with the article. It must therefore be completed in full sentences and contain the exact wording you wish to be published.

2) If any authors received a salary from any of your funders, please state which authors and which funders..

**Reviewers' comments:**

Reviewer's Responses to Questions

**Comments to the Authors:**

Reviewer #1: This submission to PLoS Genetics is an excellent combination of two analyses on the white-yellow color variation of two bumblebee morphs. The first part uses mainly GWAS to identify a transcription factor as a locus driving white-yellow switches. The second part uses next-generation sequencing of a mosaic gynandromorph to examine the genomic composition of different patches of tissues (including in parts that differ at both sexual and autosomal traits).

I found this work exciting and easy to read and follow through the text and figures.

The combination of GWAS and gynandromorph analysis is really original and fascinating, and I recommend this article for publication in PLOS Genetics, pending the authors can address the few points below.

---PART 1 (yellow-white switch)---

- It would be helpful to ensure the naming of BarH2 is correct.

I'd suggest including a molecular phylogeny of BarH2 orthologous proteins from various Holometabola insects (or more, see below).

I recommend using "ClipKIT in the browser" or other tools remove spurious sites, and IQ-TREE

It would also a good opportunity to clarify when B-H1 and B-H2 copies evolved. From a quick search I could not find reliable information on whether if copies are present outside of Diptera, for example.

- l.23 : "indicates co-mimicking species use different mutations for their white-yellow"

Consider replacing "mutations" by "cis-regulatory variations" or "haplotypes" for accuracy here. The use of mutation here is a semantic shortcut that can be appropriate in other contexts.

- Entirely up to the authors, but it would be useful to name the color locus, and use capitalization to refer to its Dominant/recessive alleles throughout.

For example

"White thorax (Wth)"

or "Nivea (Niv)" locus

Niv/- - white

niv/niv - yellow

Or B-H2 Superscript (Niv/-) vs B-H2 Superscript(sul/sul) to maintain the sub-species terminology but frame these as BarH2 alleles.

This could notably help with the text of the gynandromorph section.

On the locus name, there are a myriad of possible names, and maybe something borrowed from turkish culture would work here. I remember recent color loci with mythological references from groups in Finland (Valkyrie locus) and China (Little Red Boy). Could be an inspiration.

- a very minor stylistic comment : The section headers of the Results part read like Methods, e.g. "PCR and Sanger Sequencing". I'd suggest more informative headers to help the readers.

- in Figure 1, for clarity, it would be helpful to clarify where "vor" is on the tree (as a form of niveatus...), even if this info is elsewhere.

- in Figure 1, I would find it amazing if the authors ALSO included an additional map + pictures of yellow morphs, to more intuitively lead into Figure 2. For instance, from left to right top to bottom:

Yellow map ---- Yellow morphs

White map ----- White morphs

Phylo Tree

- Figure 2B,C: it would be helpful to feature the full names of the morphs

- Figure 3 and the GWAS are well made. Would it be possible to clarify whether heterozygotes were found from these data? Also, I think I understand why the coverage drops gradually in this region (V shape in vor), but it would be useful to feature read-pile ups (BAM alignments in IGV?) to visualize the mentioned "sharp drop in coverage", due to lack of anchoring in the indel part.

- l.205-208 "Heterozygosity Analysis". PCR/Sanger data were not shown, but suggests that two "white workers were heterozygous for the duplication". I'd expect, clean, solid evidence for this and if possible on more replicates in the revised version of this manuscript.

- Figure 4: before A-C, I would appreciate seeing how similar the niv segmental duplicates are. An alignment and maybe a dotplot of the two recovered alleles could help characterizing this SV, among other things.

---PART 2 (gynandromorph)---

I appreciate the quality of the work but the results are complex, and I had trouble to grasp them fully or to really understand if the conclusions are supported.

MAJOR SUGGESTIONS:

- Figure 5 and S5 are not color-blind friendly. For example I'd suggest picking more complementary colors in Fig. 5 (opposite on the color wheel, e.g blue and orange). Fig S5 is particularly bad in that regard and needs a rethink.

- are the authors convinced that one of the haploid source is female as in Fig S5D?

In honeybees, a study suggested that supernumerary sperm can generate one or more haploid lineage (10.1098/rsbl.2018.0670).

I tried two wrap my head around this :

307 Using heterozygous sites in the diploid RSA tissue as a reference, we compared

308 the haploid LLA and RPW samples to the RSA diploid sample and found that the

309 majority of heterozygous sites present in the diploid individual differed between haploids

310 (98.25% of 33,187), thus these mostly represent a split of the chromosomal variants of

311 the diploid. Phasing also suggested that haploids equally split the phasing patterns of

312 the diploid. However, 12.7% of the sites that differ between the haploids (n=4925) are

313 not in the diploid, suggesting that the parental origin of the haploids is not simply the

314 diploid tissue (S5C Fig.).

If I interpret this correctly, LLA and RPW are both haploid but distinct.

Fig S5D suggests n2 is derived from a polar body.

10.1098/rsbl.2018.0670 suggest it could be sperm-derived instead, assuming polyandrous matings occur.

I'd love more clarity on this and proper visualization of these different models.

We are likely to see more studies of this type in gynandromorphs in the future.

- if the gynandromorph is a mix of more than two parental haplogenomes, it would be useful to visualize multi-allelism over selected loci, with alignments or BAM read-piles.

For example I'd be curious to see the Bar-H2 allelic states over at least LLA RPW and RSA.

Or if the phasing worked as the authors state, maybe some regions of the genomes show clear evidence for the presence of 4 alleles.

MINOR SUGGESTIONS

-l. 235 : year of collection? (relevant to inspire future analysis of this type")

Or more specimen information could be added to the Methods section.

- l.283 "the simplest phenotype-based explanation is somatic loss of one chromosome set in a subset of cells, producing haploid (male) cells on the yellow side with the deletion allele, and diploid (female) cells on the white side that are heterozygous for the color locus".

I disagree, the simplest explanation is genome-wide chimerism with a mix of diploid and haploid. NOT mosaicism in aneuploid sex determination locus AND color locus, since these are likely to be UNLINKED.

I'd recommend rephrasing and clearly stating the major hypothesis based on these phenotypes, including the mention that the color locus and Sex Determination locus are on separate chromosomes (unless I missed something?)

- Figure 5 : location of sequenced tissues? I found that the use of acronyms made it difficult to follow the analysis. A visualization would help, and the use of short/long antennae may not be as intuitive as their male/female status. The pairwise design with female/male antennae and yellow/white pleuron is neat. This was completely lost in the figures.

- Figure S5 : panel number order is atypical (A D B C)

- Figure S5A : what is the state of the color locus in the reference genome used here?

The legend mentions proportions of non-reference alleles.

- Figure S5B and C : these data are difficult to tackle, any visual aid would be appreciated. Even spending time time on the legend and text, I could not fully understand what's shown here.

Reviewer #2: PLOS Genetics manuscript "The genetic basis of mimicry in the snowy bumble bee (B. niveatus) in Anatolia with insights from a color polymorphic gynandromorph" (PGENETICS-D-25-01242)

The authors have conducted a genomic analysis of a bee color polymorphism, finding evidence that allelic variation near the BarH2 region is causally associated with the formation of a white morph, which appears to be unique to this species and not shared with other taxa in a mimicry ring. Overall, I find the evidence compelling for these general findings.

While I would have liked to see additional data (transcriptomics, histology) to support their causal insights, I will limit my suggested additional work to the data they have in hand. I am concerned about reference bias in their analyses and suggest that they query their genomic data for ways to try and resolve these issues. Since the implicated region appears to be free from repeat content, I suggest they try to generate genome assemblies using their GWAS data, independently by morph, to assembly the alternative haplotypes, and gain more insight into why their read depth variation is being observed. My hunch is that this is arising due to reference bias. Another way to assess this would be to conduct a kmer based anlaysis of their samples (e.g. see Willink, et al (2023). The genomics and evolution of inter-sexual mimicry and female-limited polymorphisms in damselflies. Nat Ecol Evol). Using their existing genomic data to better resolve the white vs yellow allele in this region could greatly improve their genomic signatures and thereby causal hypotheses.

Specific comments:

If B. niveatus and B. vorticosus are now recognized as the same species, why do the authors continue to use their original species level distinction? Fixing this throughout the text would be preferred.

Read depth drop in B. vorticosus (L156-158) suggests need for a kmer reference free approach to sensure that all causal regions are identified without reference bias effects. I assume your reference was for white morph?

What is the read depth profile being shown in Fig3E? is this for a representative individual from each, or average across individuals? If this is for single individuals of each morph, what about including multiple to convey sense of variation across individuals, and consistency between morphs? Fig3F is not clear … what is being shown? How is this polaraized? Please add additional details to the figure and caption.

L174-6: would have been better to sequence additional individuals for SNP-trait assocaitions.

L203. This was an important step, great. But again here you are using subspecies designation, but they are just morph differences? Again, this goes back to the early critique I had with regards to terms and latin names. If you from the state just call them by morph, being the same species, that would help. Here because you are saying subspecies, it leads me to wonder what you are sampling. With the white allele dominant, then how many of your individuals in the GWAS were hets / vs. homozygous? Does that help with assessing your read depth issues?

Figure quality/resolution is low throughout.

Fig1.

Species names in upper right are not easily visible. What are the white/yellow circles in on the phylogeny.

Fig2B logic comes from the GWAS, but here it is being shown before the GWAS results are presented, which is a bit problematic (i.e. lines L150).

**Have all data underlying the figures and results presented in the manuscript been provided?**

Reviewer #1: None

Reviewer #2: Yes

PLOS authors have the option to publish the peer review history of their article (what does this mean? ). If published, this will include your full peer review and any attached files.

**Do you want your identity to be public for this peer review?** For information about this choice, including consent withdrawal, please see our Privacy Policy .

Reviewer #1: No

Reviewer #2: No

**Figure resubmission:**
---

## [Decision Letter · Decision Letter 1]

16 Feb 2026

Dear Dr Hines,

We are pleased to inform you that your manuscript entitled "The genetic basis of mimicry in the snowy bumble bee (B. niveatus) in Anatolia with insights from a color polymorphic gynandromorph" has been editorially accepted for publication in PLOS Genetics. Congratulations!

Yours sincerely,

Artyom Kopp

Academic Editor

PLOS Genetics

Justin Fay

Section Editor

PLOS Genetics

Aimée Dudley

Editor-in-Chief

PLOS Genetics

Anne Goriely

Editor-in-Chief

PLOS Genetics

BlueSky: @plos.bsky.social

Comments from the reviewers (if applicable):

Reviewer's Responses to Questions

**Comments to the Authors:**

Reviewer #1: The revisions improved the manuscript with better clarity, and the limitations of the results are carefully discussed.

Reviewer #2: I have now had a chance to read through the responses to reviewer comments. I find the authors have thoughtfully responded to both the reviews and the editor. I have also read through their revised text sections. I find the revision to be well conducted and am satisfied.

**Have all data underlying the figures and results presented in the manuscript been provided?**

Reviewer #1: Yes

Reviewer #2: Yes

PLOS authors have the option to publish the peer review history of their article (what does this mean? ). If published, this will include your full peer review and any attached files.

**Do you want your identity to be public for this peer review?** For information about this choice, including consent withdrawal, please see our Privacy Policy .

Reviewer #1: No

Reviewer #2: No

**Data Deposition**

http://datadryad.org/submit?journalID=pgenetics&manu=PGENETICS-D-25-01242R1

**Press Queries**

---

## [Editor Report · Acceptance letter]

PGENETICS-D-25-01242R1

The genetic basis of mimicry in the snowy bumble bee (Bombus niveatus) in Anatolia with insights from a color polymorphic gynandromorph

Dear Dr Hines,

We are pleased to inform you that your manuscript entitled "The genetic basis of mimicry in the snowy bumble bee (Bombus niveatus) in Anatolia with insights from a color polymorphic gynandromorph" has been formally accepted for publication in PLOS Genetics! Your manuscript is now with our production department and you will be notified of the publication date in due course.

With kind regards,

Lilla Horvath

PLOS Genetics

On behalf of:
